# Projected Changes in the Atmospheric Dynamics of Climate Extremes in France

**Pascal Yiou** [1],*, **Davide Faranda** [1,2,3], **Soulivanh Thao** [1] **and Mathieu Vrac** [1]

1   Laboratoire des Sciences du Climat et de l'Environnement, UMR8212 CEA-CNRS-UVSQ,
    U Paris-Saclay & IPSL, 91191 Gif-sur-Yvette, France; davide.faranda@lsce.ipsl.fr (D.F.);
    soulivanh.thao@lsce.ipsl.fr (S.T.); mathieu.vrac@lsce.ipsl.fr (M.V.)
2   Laboratoire de Météorologie Dynamique, Ecole Normale Supérieure, PSL University & IPSL,
    75005 Paris, France
3   London Mathematical Laboratory, 8 Margravine Gardens, London W6 8RH, UK
*   Correspondence: pascal.yiou@lsce.ipsl.fr

**Abstract:** Extremes of temperature, precipitation and wind have caused damages in France, in the agriculture, transportation and health sectors. Those types of events are largely driven by the atmospheric circulation. The dependence on the global climate change is not always clear, and it is the subject of extreme event attribution (EEA). This study reports an analysis of the atmospheric circulation over France for seven events that struck France in the 21st century, in various seasons. We focus on the atmospheric dynamics that leads to those extremes and examine how the probability of atmospheric patterns and their predictability responds to climate change. We analyse how the features of those events evolve in simulations following an SSP585 scenario for future climate. We identify how thermodynamical and dynamical changes of the atmosphere affect the predictability of the atmospheric circulation. Those using a range of CMIP6 simulations helps determining uncertainties linked to climate models.

**Keywords:** extreme events; climate change; France





## 1. Introduction

Rather than a global perspective, decision-makers have emphasised the necessity of projections of regional extremes for adaptation to climate change [1]. Therefore, this paper will focus on a few extreme climate events that recently occurred in continental France. Insurance, health, agriculture and energy sectors (at least in France) are often affected by thermal extremes (cold and hot), extreme precipitation or wind speed. Observations show that the duration of such events rarely exceeds a couple of weeks for a maximum impact. Therefore, the key time scale we will consider in this paper is sub-seasonal.

A climate or meteorological extreme event is often defined when a key variable (temperature, precipitation, wind speed) exceeds a predefined threshold. This allows computing the probability of the event, and then the change of probability of that event, conditional to a climate change scenario, which is the essence of extreme event attribution (EEA) [2–4]. Many papers of EEA are based on the estimate of probabilities of events, from statistical modelling of exceeding a threshold. This requires tools from extreme value theory (EVT) [5]. The main caveat of some of those studies is that they do not take into account the physical processes leading to the extreme events, like features of the atmospheric circulation. Attempts to connect the atmospheric circulation to variables that define the extremes have been recently devised [6,7]. Shepherd [8] argued that the atmospheric circulation is a key element of the uncertainty in attribution studies. This motivates our focus on features of the atmospheric variability that drive a few key events.

New mathematical tools of extreme event attribution have been recently devised [9]. These tools focus on dynamical properties of the atmospheric circulation. Dynamical

properties of physical systems correspond to time derivatives of the variables of the system, which can be determined from well-chosen mathematical and statistical indicators [10–12]. Faranda et al. [10] have argued that such dynamical indicators are related to the predictability of the (atmospheric) system. Therefore, our paper will deal with estimates of dynamical features linked to the atmospheric predictability. The dynamical indicators we will consider include the local dimension, persistence and pattern likelihood.

In this paper, we study the atmospheric circulation that prevailed during several extreme climate events that occurred in France since 2000. Our goal is to determine how the atmospheric motion properties can be altered by climate change.

Without being exhaustive, we decided to have a wide panel of types of events. Therefore, the events we consider are based on temperature, precipitation or wind speed, and which occurred in the four seasons. In the mid-latitudes, those quantities are linked to the large-scale atmospheric circulation [13]. This justifies a focus on the atmospheric dynamics that is related to the considered extremes. Therefore, we shall examine how the synoptic scales of the atmosphere that occur during regional temperature, precipitation or wind events can be affected by climate change.

We will determine the values of dynamical indicators of the atmospheric circulation in a reanalysis dataset, for reference extreme events. Then, we will assess how those dynamical indicators during those reference events will change, as a response to a future climate scenario (here SSP585). This will be performed by sampling the atmospheric variability due to future climate change from a multi-model ensemble (the Coupled Model Intercomparison Project, phase 6: CMIP6 [14]). Before performing such a task, a bias adjustment method will be applied to the modelled CMIP6 fields of geopotential height at 500 hPa (Z500), in order to remove their main biases with respect to a reanalysis dataset. To do so, two experiments will be conducted. The first one will directly correct the modelled Z500, while the second one will first detrend the Z500 fields before applying the bias adjustment. The detrending corresponds to removing the spatially uniform shift of Z500, mostly caused by the warming over the region. Thus, by detrending, we will remove the first-order thermodynamic effect of warming on Z500 fields. This will allow the bias correction method to be applied on anomalies to focus on changes essentially due to dynamical changes of the state of the atmosphere. This procedure allows comparing the relative contributions of thermodynamical and dynamical changes of those atmospheric features.

Model and reanalysis data will be described in Section 2. The selection of events is described in Section 3. Methods will be exposed in Section 4. Results appear in Section 5. Discussion and conclusion appear in Section 6.

## 2. Data

### 2.1. Observations and Reanalyses

We considered the ERA5 reanalysis [15] for the determination of key extreme events since 2000. The ERA5 period is available from 1979 to now, with a horizontal resolution of 0.25°. We computed daily averages for sea-level pressure (SLP), geopotential height at 500 hPa (Z500), 2 m temperature (T2m), total precipitation (Pr, which combines rainfall, snowfall and hail), 10m wind speed and 10 m peak wind speed (W, Wmax). SLP and Z500 fields were extracted for a region covering 80 W–30 E and 30 N–65 N. Temperature, precipitation and wind speed fields were extracted over a region covering France (4.5 W–8.5 E; 42 N–51.5 N). We then selected the 1018 gridpoints that are included in continental France (excluding Corsica).

### 2.2. Climate Model Simulations

We analyse daily output of the Coupled Model Intercomparison Project phase 6 (CMIP6) [14] for 11 historical simulations (see Table 1 for references), and one socio-economic pathway (SSP) scenario [16]. This selection was dictated by the availability of Z500, temperature and precipitation fields on daily time scales at the time of analyses: we have only selected models whose data were fully available for the whole period 1950–2100.

The historical simulations cover the period 1950–2014. The forcings are consistent with observations. They include changes in atmospheric composition due to anthropogenic and volcanic influences, solar forcing, emissions or concentrations of short-lived species and natural and anthropogenic aerosols or their precursors, as well as land use. The SSP585 scenario corresponds to a representative concentration pathway scenario (RCP) with a radiative forcing increase of 8.5 Wm$^{-2}$ in 2100 due to greenhouse gas emissions, relative to pre-industrial conditions [16].

**Table 1.** List of CMIP6 simulations used in this study, their approximate horizontal resolution and references.

| Simulation Name | Atmospheric Resolution | Data Reference |
|---|---|---|
| BCC-CSM2-MR | 100 km | [17] |
| CanESM5 | 500 km | [18] |
| CNRM-CM6-1-HR | 100 km | [19] |
| CNRM-CM6-1 | 250 km | [20] |
| CNRM-ESM2-1 | 250 km | [21] |
| INM-CM4-8 | 100 km | [22] |
| INM-CM5-0 | 100 km | [22] |
| IPSL-CM6A-LR | 250 km | [23] |
| MIROC6 | 250 km | [24] |
| MRI-ESM2-0 | 100 km | [25] |
| UKESM1-0-LL | 250 km | [26] |

The simulations of each climate model came in ensembles of several members. For simplicity and because not all ensemble sizes were equal, we picked one member of each model ensemble. This is a caveat of this study, but this avoids a complicated discussion on weighing models by the provided ensemble size.

## 3. Selection of Events

We decided to chose extreme events that hit France since the beginning of the 21st century. This section details a selection criterion based on a combination of intensity, duration and spatial extent of the climate variable that was extreme (temperature, precipitation or wind speed). We then describe the meteorological conditions that prevailed during each event and its impacts.

We determined key years for extremes of temperature, precipitation and wind that occurred in the 21st century. We used a simplified version of the approach of Cattiaux and Ribes [27]. We first considered each season separately (Winter: December-January-February, DJF; Spring: March-April-May, MAM; Summer: June-July-August, JJA; Autumn: September-October-November, SON), as the features of extremes depend on the season. For each season and each grid point of the ERA5 reanalysis, we determined the 5th ($q_{05}$) and 95th ($q_{95}$) quantiles of near-surface temperature (T2m), total precipitation (Pr), wind speed (W) and maximum wind speed (Wmax), between 1979 and 2020. Then, for each day $t$ and year $y$ and each grid point $s$, we determine whether the value of climate variable $X \in \{T2m, Pr, W, Wmax\}$ exceeds a 95th quantile threshold. The reason for this procedure is the high dependence of precipitation and wind speed on altitude: wind speeds exceeding 100 km/h or precipitations rates over 100 mm/day are not particularly extreme in mountainous areas, while they are rather rare in plains. Therefore, it is necessary to consider location dependent thresholds (i.e., local quantiles) to detect extremes.

$$\pi(s,t,y) = \left\{ \begin{array}{ll} 1, & \text{if } X \geq q_{95} \\ 0, & \text{if } X < q_{95}. \end{array} \right. \tag{1}$$

For each year $y$, we compute the empirical probability that $X$ exceeds a high threshold ($q_{95}$) (or is below a low threshold ($q_{05}$)):

$$p(y) = \frac{1}{N_{FR}N_{season}} \sum_s \sum_{t \in y} \pi(s,t,y), \qquad (2)$$

where $N_{FR}$ is the number of grid point in France ($N_{FR} = 1018$ in ERA5) and $N_{season}$ is the number of days in a season ($90 \le N_{seas} \le 92$). Here, $N_{seas}$ in winter can vary during leap years. This quantity accounts for the intensity of events (the variable $X$ has to exceed a high threshold), spatial extent (fraction of grid cells for which $X$ exceeds a threshold) and duration (number of days for which $X$ exceeds a threshold).

Figure 1 shows the time series of $p(y)$ for temperature, precipitation and wind speed. The higher values for temperature extremes (Figure 1a, vertical axis) reflect that thermal events have a larger geographical extent and a longer duration (2–3 weeks). Storms travel across France in a couple of days. Precipitation events generally cover very limited areas and last a couple of days.

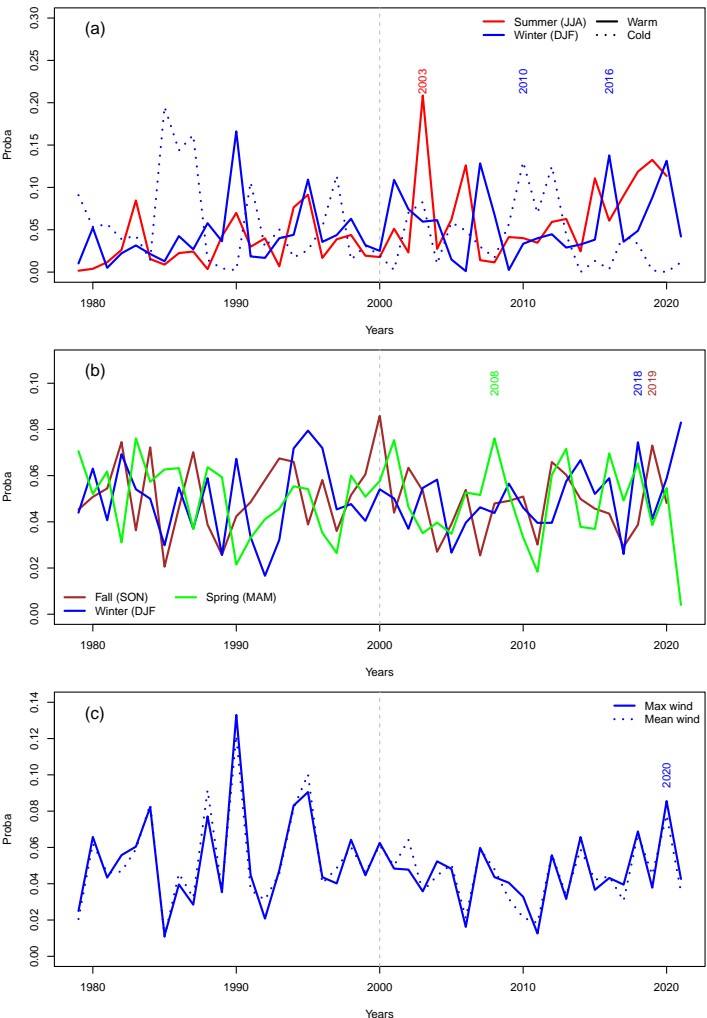

**Figure 1.** Variations of $p(y)$ for temperature (**a**), precipitation (**b**) and wind speed (**c**). The vertical dashed line is for the year 2000 after which the events are considered. The colours indicate the seasons: red for summer, brown for Autumn, blue for winter and green for spring. The dotted line in panel (**a**) indicates the value of $p$ when temperature is below the 5th quantile. The years indicate the records during the 21st century, but excluding 2021.

In this paper, in order to avoid a tedious exhaustive list of events, we focused on warm winters and summers (which have impacts on human health and ecosystems), cold winters (which have impacts on energy demand and human health), wet springs (that have impacts on agriculture and river management) and autumns (with Mediterranean events and subsequent flash floods), and stormy winters (which can be destructive).

With this approach, we outline seven remarkable recent years (since 2000), with examples of warm summer (2003) and winter (2016), cold winter (2010), wet spring (2008), wet winter (2018), wet autumn (2019) and windy winter (2020). Then we identified event dates from (daily) time series of climate variables. We determined clusters of days when the considered climate variable $X$ exceeds the seasonal average of the 95th quantile of 1018 geographical points (in continental France), with possible excursions below this threshold of no more than 2 days. Shorter "excursion" times to less extreme states (i.e., 1 day or a fully continuous cluster) could have been considered, depending on the impacts of the event. For example, the insurance sector considers that events must be separated by at least 24 h to be treated separately. However, this only marginally affects the analyses presented in this paper.

The features of those seven identified events are summarised in Table 2.

**Table 2.** Table of key extreme events in France since 2000. The median local dimension $d$ and persistence metric $\theta$ (see Section 4 for definitions) are indicated for each event.

| Date of Event | Cold | Warm | Wet | Storm | $d$ | $\theta$ |
|---|---|---|---|---|---|---|
| 31 July 2003–17 August 2003 | | X | | | 10.64 | 0.34 |
| 09 March 2008–01 April 2008 | | | X | | 10.06 | 0.43 |
| 11 December 2009–18 February 2010 | X | | | | 9.27 | 0.37 |
| 01 December 2015–13 January 2016 | | X | | | 9.70 | 0.44 |
| 25 December 2017–09 January 2018 | | | X | | 9.83 | 0.45 |
| 13 October 2019–25 October 2019 | | | X | | 11.63 | 0.46 |
| 06 December 2019–15 December 2019 | | | X | X | 9.75 | 0.48 |

*3.1. Summer Heatwaves*

The summer 2003 was a heatwave epitome, in term of amplitude and duration [28,29]. This events had huge impacts on the biosphere [30], public health [31] and the economy [32,33].

As of 2021, the mean summer temperature of 2003 in France is still the all time record since the beginning of meteorological records.

The mean temperature reached its climax between 31 July and 17 August 2003. The mean seasonal temperature anomalies are shown in Figure 2 (left). The event was preceded by a drought that started in early May 2003. This prolonged drought hindered latent heat fluxes and hence exacerbated high temperatures. The heat anomaly was also characterised by tropical night time temperatures.

The atmospheric circulation was characterised by a strong anticyclonic pattern centred over France (Figure 2 (right)), during which almost no wind blew over France, which also enhanced ozone air pollution (and the associated death toll).

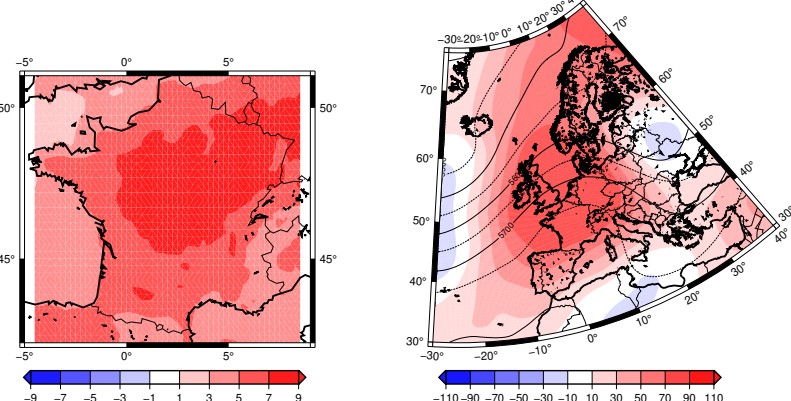

**Figure 2. Left panel**: mean anomalies of T2m over France between 31 July 2003 and 17 August 2003. **Right panel**: mean anomalies of Z500 (in m) over the East North Atlantic region (colours). Contour lines indicate the mean Z500 (in m).

### 3.2. Winter Cold Spells

The winter of 2009–2010 was exceptionally cold and snowy in Europe and Eastern US [34] (Figure 3 (left)). Although the temperatures were not as cold as historical events such as the winters 1954 or 1963, this cold event created havoc in transportation systems, and was deemed to be the coldest since the beginning of the 21st century.

The atmospheric circulation was cyclonic, with a persisting negative phase of the North Atlantic Oscillation (Figure 3 (right)) [34,35].

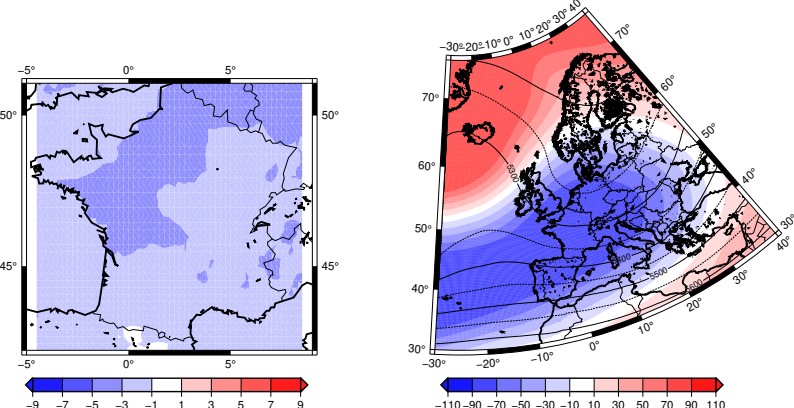

**Figure 3. Left panel**: mean anomalies of T2m over France between 11 December 2009 and 18 February 2010. **Right panel**: mean anomalies of (in m) over the East North Atlantic region (colours). Contour lines indicate the mean Z500 (in m).

### 3.3. Winter Warm Spells

The winter of 2015–2016 was particularly warm in France (Figure 4 (left)). It also corresponds to an exceptionally strong El Niño [36]. The previous record of winter temperature occurred in 2006–2007 [37]. This warm spell had a large geographical extent in the northern hemisphere [38,39]. In December 2015, no negative temperatures (in Celsius) occurred in France. This had consequences on phenological cycles of plants that require freezing temperatures in order to build defences against pests [40].

The atmospheric circulation had an anticyclonic pattern over France that extended into North Africa (Figure 4 (right)). This warm episode was responsible for air pollution due to atmospheric stagnation [41] in December 2015.

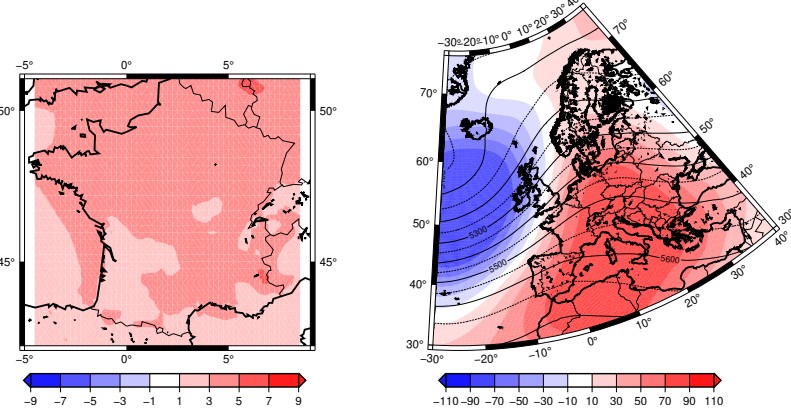

**Figure 4. Left panel**: mean anomalies of T2m (in K) over France between 1 December 2015 and 13 January 2016. **Right panel**: mean anomalies of Z500 (in m) over the East North Atlantic region (colours). Contour lines indicate the mean Z500 (in m).

### 3.4. Wet Spring Events

Most of March 2008 was rainy. The daily maxima were not exceptional, but the cumulated precipitation flooded northern and central France.

The atmospheric circulation yielded a persisting cyclonic pattern (Figure 5 (right)) that conveyed moist air into France.

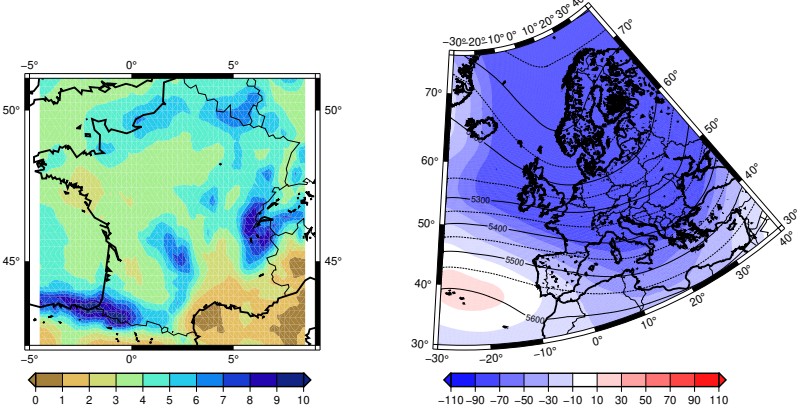

**Figure 5. Left panel**: mean precipitation rate (in mm/day) over France between 9 May 2008 and 1 April 2008. **Right panel**: mean anomalies of Z500 (in m) over the East North Atlantic region (colours). Contour lines indicate the mean Z500 (in m).

### 3.5. Wet Winter Events

Southeastern France witnessed short and intense precipitation at the beginning of December 2017 (storm Ana). The end of December 2017 had a longer spell of precipitation, which hit most of France (Figure 6 (left)), due to a spate of storms (named Bruno, Dylan, Carmen and Eleanor).

The atmospheric circulation yielded a zonal pattern (Figure 6 (right)) that brought intense and prolonged precipitations in the southern half of France, along with the storms [42].

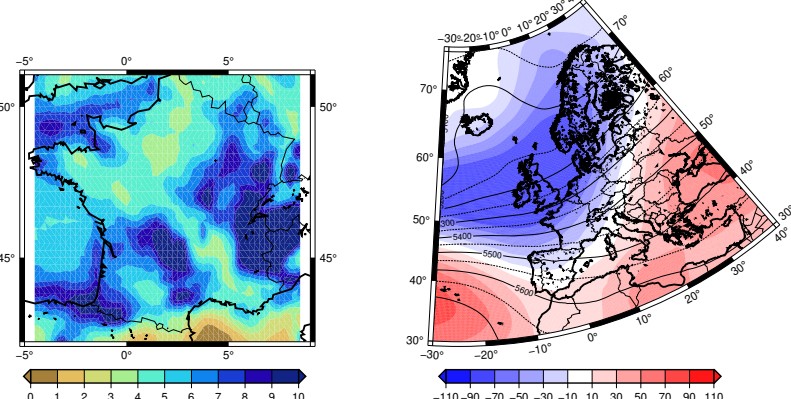

**Figure 6. Left panel**: mean precipitation rate (in mm/day) over France between 25 December 2017 and 9 January 2018. **Right panel**: mean anomalies of Z500 (in m) over the East North Atlantic region (colours). Contour lines indicate the mean Z500 (in m).

### 3.6. Autumn Mediterranean Events

Mediterranean precipitation events occur in the Autumn season over the mountainous regions of the Mediterranean arc, when the Mediterranean sea is still warm and high altitude air has cooled down. Such conditions create strong convective events with devastating effects.

The Autumn of 2019 witnessed a large number of Mediterranean events, with a climax in October 2019, during which the Aube region (south east of France) witnessed catastrophic floods that lead to many casualties.

During the October events, the atmospheric circulation had the conjunction of a cyclonic pattern in the eastern North Atlantic and an anticyclonic pattern over central Europe (Figure 7 (right)). This meridional circulation pumped moisture into southern France, with a large amount of precipitation during almost two weeks.

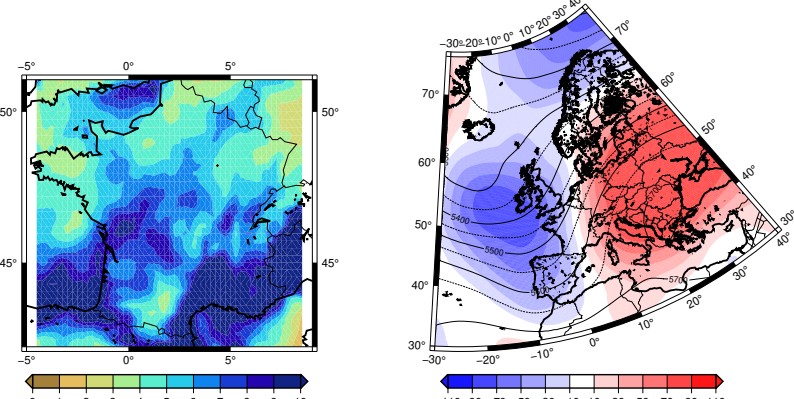

**Figure 7. Left panel**: mean precipitation rate (in mm/day) over France between 13 October 2019 and 25 October 2019. **Right panel**: mean anomalies of Z500 (in m) over the East North Atlantic region (colours). Contour lines indicate the mean Z500 (in m).

### 3.7. Winter Storms

Winter North Atlantic storms occur every winter. They start near the Labrador Sea and move across the North Atlantic, following the storm track [43,44]. A few of them can hit France or Europe, especially when the atmospheric circulation is zonal.

The winter of 2019/2020 was particularly stormy, with 10 named storms that hit France between December 2019 and February 2020. Three named storms (Atiyah, Daniel and Elsa) hit France between December 6th and December 12th 2019, with wind speeds

exceeding 150 km/h. During that week, the atmospheric circulation was zonal (Figure 8 (right)), with cyclonic conditions over Iceland, and anticyclonic conditions near the Azores.

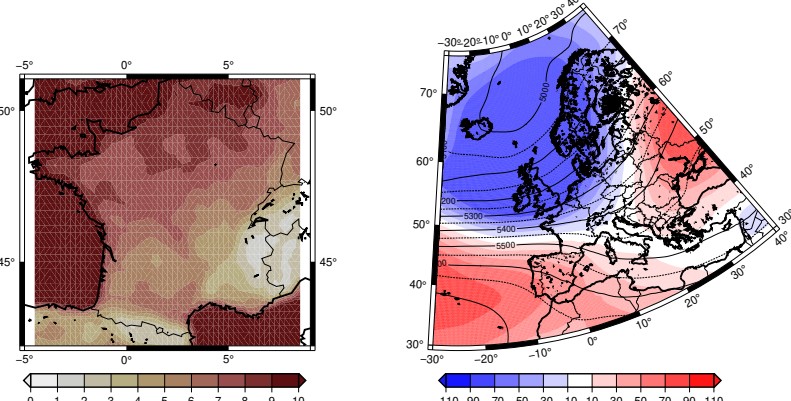

**Figure 8. Left panel**: mean maximum daily wind speed (in m/s) over France between 6 December 2019 and 15 December 2019. **Right panel**: mean anomalies of Z500 with respect to the seasonal cycle (in m) over the East North Atlantic region (colours). Contour lines indicate the mean Z500 (in m).

This spate of storms also brought a lot of precipitation over France, and caused local floods.

## 4. Methods

### 4.1. Bias Correction and Trend Removal

Given that the models have biased representation of the Z500, we apply a statistical bias correction [45] on the Z500 fields allowing to account for climate change [46].

The statistical bias correction method applied is the Cumulative Distribution Function-transform (CDF-t) method, developed by Michelangeli et al. [45]. The bias corrections were made through the CDF-t R package (available at https://cran.r-project.org/web/packages/CDFt accessed on 29 October 2021). More theoretical and technical details, as well as first validations and comparisons can be found in [9,46].

This approach links the cumulative distribution function (CDF) of a climate variable (here Z500) from GCM simulations to be corrected, to the CDF of this variable from a reference dataset, here the ERA5 reanalysis dataset [15]. CDF-t can be considered as a variant of the empirical quantile-mapping method but within the appropriate target (here future) time period and therefore accounts for changes of CDF from the calibration period to the projection one.

This bias correction method is applied for each grid-point separately in two different ways. First, CDF-t is applied on a monthly basis to the "raw" ERA5 reanalyses and 11 CMIP6 GCM simulations. The results are called the "non-detrended bias corrections". The effect of Z500 bias correction on the mean values and standard deviation (in December and August) are illustrated in Figures A1 and A2.

Second, CDF-t is applied (also on a monthly basis) to ERA5 reanalyses and GCM simulations from which a spatial and seasonal trend is removed. To do so, for each day (in ERA5 and CMIP6), the Z500 spatial average is calculated. Next, for each calendar day (e.g., each January, 1) over the periods of interest (1979–2019 or 2061–2100), a linear fit of the daily Z500 spatial average as a function of time is estimated. This spatial trend is then removed from each Z500 grid-cell value for the specific calendar day. Then, the spatial average value estimated for the model during the year 2000 is added to the calendar day. This ensures that a seasonality (estimated for 2000) is preserved, with no trend in the resulting Z500 data. Those seasonally and spatially detrended data are the inputs of CDF-t, providing adjusted values. Hereafter, we refer to those adjusted values as "detrended bias corrections". The removed Z500 spatial average trend corresponds to the spatially uniform shift of Z500, mainly caused by the warming over the region. Therefore, by removing this

trend, we also removed the first-order thermodynamic effect of warming on Z500 fields. Therefore, the resulting anomalies, which are further bias corrected, indicate changes that are mostly due to dynamical changes of the state of the atmosphere.

CDF-t bias correction can be applied on detrended and non-stationary data [46–50]. Indeed, unlike other bias correction methods that require stationary distributions, the CDF-t approach is explicitly designed to account for changes in the distributions, i.e., to account for non-stationarity, and is thus suited when trends are present in data. This does not imply that CDF-t corrects the trends of the model, but rather that CDF-t mostly preserves the trends from the model data to be corrected.

### 4.2. Variations of Dynamical Indicators due to Climate Change

We extract the daily Z500 fields corresponding to the selected extreme event in the ERA5 reanalysis. We then embed these observed trajectories into historical simulations (1979–2019) and projections (2061–2100) under a high (SSP585) emissions scenarios of the CMIP6 models [14]. This embedding of observed trajectories in climate simulations is performed by looking at best analogues. Using the extreme event attribution vocabulary, we consider the historical simulations (1979–2019) as the *factual* world, i.e., the actual world with the current level of anthropogenic emissions. The SSP585 scenario corresponds to a *counterfactual* world. Contrary to the traditional counterfactual world representing the world that could have been without climate change, our counterfactual world explores the world that could be under projected trajectories of anthropogenic emissions [2]. This *forward* EEA approach has been used by Van Oldenborgh et al. [51], who calculated trends up to 2100 from model outputs, Sweet et al. [52] who evaluated the annual maximum storm tide level for four different scenarios of sea level rise (see also Sweet et al. [53]) and by Kay et al. [54], Yoon et al. [55] to project the evolution of fire risks in the future. When we embed the portion of Z500 trajectories corresponding to extreme events, we assume that the circulation patterns associated with the extreme event could be observed in the climate model simulations. This assumption is justified by previous studies where it has been verified that the average analogue distances between observed atmospheric patterns (for Z500 and sea-level pressure) and the historical model simulations are within the analogue distance of the ERA5 reanalysis [56].

#### 4.2.1. Analogues Computation

For each extreme event, we follow the approach of Faranda et al. [9] by computing the analogues of the observed synoptic patterns in each set of model simulations, and determine their properties. For each daily Z500 field observed during extreme events, we compute the Euclidean distance from all the other daily Z500 fields by a spatial average of grid-point distances and we then select the closest 2% daily Z500 fields. The Euclidean distance $D$ is computed over maps of Z500, between two days $t$ and $t'$:

$$D(t,t') = \left[ \sum_x \big( Z500(x,t) - Z500(x,t') \big)^2 \right]^{1/2}, \tag{3}$$

where $x$ spans the ensemble of grid points over the North Atlantic domain. For each day $t$, we consider the closest days $t'$, which yield values of $D$ that are smaller than the 2% quantile of all values. This defines our analogues ensemble. Note that the results do not crucially depend on this percentage provided that it is in the range of 0.5 to 3%. The values of the Euclidean distance allow to determine how well the circulation patterns associated with extreme events fit in the simulations. This metric is what we call 'analogue quality', defined as the average of the Euclidean distance of the 2% closest fields.

In addition, we compute the local dimension $d$ and a persistence $\theta$ metrics, which are linked to the predictability of the circulation (see [10,57,58]).

### 4.2.2. Local Dimension

The local dimension $d$ around a configuration $\zeta$ of Z500 (at time $t$) describes the statistics $-\log(D(t, t')$ for all times $t'$ for which $D(t, t')$ is below the 2% quantile threshold. The distribution of $-\log(D(t, t')$ generically follows an exponential distribution whose parameter $\lambda$ is linked to the local dimension through the relationship $\lambda = 1/d$ and a 0 shape parameter [11]. The local dimension informs on the local number of degrees of freedom of the state $\zeta$, i.e., the number of different states that could appear after $\zeta$. Hence a low $d$ indicates few future possibilities, and a higher value of $d$ indicates many potential choices when the climate system leaves $\zeta$.

### 4.2.3. Local Persistence

We can define a metric of persistence $\theta$ of a state $\zeta$ of Z500 by measuring the time it takes to escape the state $\zeta$ through a quantity defined in extreme value theory, namely, the extremal index [10]. When persistence is high, the extremal index $\theta$ is close to zero, the system takes a long time to leave $\zeta$ and it can be considered as predictable around $\zeta$. When persistence is low, the system is considered to be less predictable because it escapes fast from its state.

### 4.2.4. Dynamical Indicators and Atmospheric Circulation

The local dimension and persistence characterise the recurrences of a system around a state in phase space. In our case, the state is the Z500 map for a given extreme event. Values of $d$ and $\theta$ are obtained for every day in the dataset of interest. As $d$ provides information on the number of pathways the system can take to reach and leave a state of Z500 [59,60], it acts as a proxy for the system's active number of degrees of freedom around the state of interest. $0 < \theta \leq 1$ is a metric of persistence [61] of an atmospheric circulation state in time, i.e., how long the system typically stays around the state of interest. A very persistent state (i.e., with $\theta^{-1}$ close to zero) is highly stable (and therefore also highly predictable), while a very unstable state yields $\theta = 1$ and therefore low persistence. The information provided by persistence and predictability are different: persistence is related to very short term predictability, namely, the possibility of observing tomorrow a pattern which resembles the one observed today. The metric $d$ for Z500 is meant to extend towards subseasonal to seasonal scales, as this metric is linked with the underlying local Lyapunov exponents of the systems [11].

The $d - \theta$ parameters are computed for each selected event, in all models and scenario runs (historical and SSP585), so that we can detect changes in the atmospheric circulation observed during extreme events. A change in the analogues quality tells whether the atmospheric configuration is more or less likely in the historical than in the scenario experiments. A change in the dynamical indices informs on the change of predictability and persistence of the circulation pattern associated with the extreme event. The probability distribution of those parameters for each event and each dataset (ERA5 and CMIP6) are shown in Figures A3 and A4.

## 5. Results

We show how the dynamical features (dimension $d$ and persistence metric $\theta$) of the atmosphere for the seven emblematic events change in scenario simulations.

The present-day values of those dynamical parameters are shown in Figure 9, obtained with the ERA5 reanalysis (1979–2019). We observe that the different events occupy preferential regions of the $d - \theta$ diagram. In particular, Z500 states associated with the cold spell of 2010 have below average $\theta$ values (high persistence) and below-average dimension. This region of the $d, \theta$ diagram was associated with the negative phase of the North Atlantic Oscillation (compare the NAO pattern in Figure 2 in [10] with Figure 3 right panel). The 2019 wet events are associated with Z500 fields displaying above average dimensions and $\theta$. This region of the $d, \theta$ diagram was associated with Blocking Pattern (compare the

BLO pattern in Figure 2 in [10] with Figure 8 right panel). The results discuss the changes of those parameters $d$–$\theta$ for a climate scenario, in CMIP6 simulations.

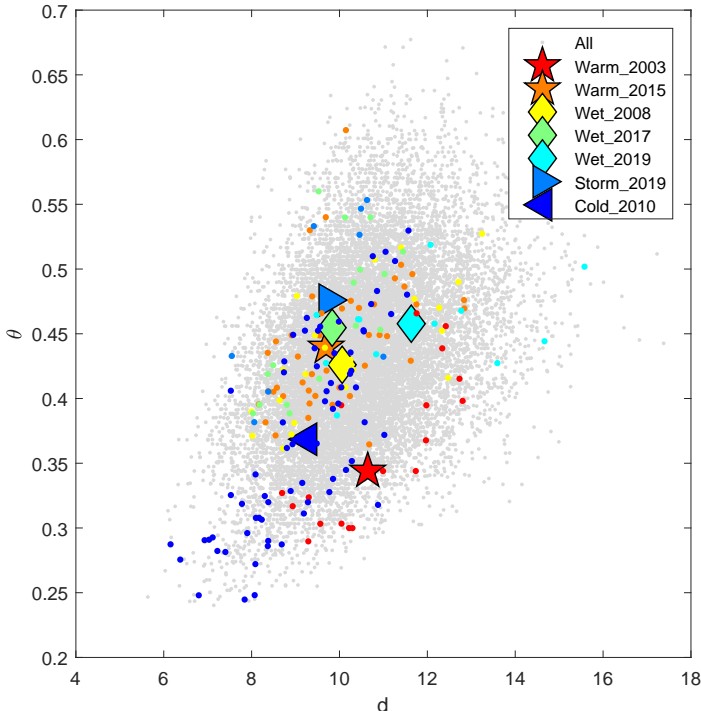

**Figure 9.** Scatter plot of the daily values of local dimension $d$ and local persistence metric $\theta$ determined from the Z500 of the ERA5 reanalysis data. The symbols indicate the median $d$ and $\theta$ values for the seven events selected in Section 3, stars for heatwaves, diamonds for precipitation events, right triangle for the 2019 storm and left triangle for the 2010 cold spell. The coloured dots correspond to daily values of $d$-$\theta$ for each identified event.

As discussed in Section 4.1, we determine $d$–$\theta$ for the seven events of the beginning of the 21st century, assuming they occur toward the end of the 21st century. We examine the influence of warming on the dynamical indicators by subtracting the trend in Z500. Therefore, the raw changes of $d$–$\theta$ indicate, to a first order, a thermodynamical contribution of climate change to the dynamics of the event. The $d$–$\theta$ changes with the detrended data corresponds to a change in circulation dynamical features related to predictability.

The results are summarised in Figure 10, which shows how the dynamical features $d$–$\theta$ change (relatively) for each event.

We emphasise that the *y*-axes are different for panels (a) and (b). This implies that keeping the trend produces stronger changes in the metrics, with up to 20% of variation (with respect to the historical period) for $d, \theta$ in the period 2061–2100 for the circulation associated with the 2003 heatwave. When removing the trends, we observe smaller variations of order 2 or 3% for all events. The impact of circulation changes is therefore smaller than the thermodynamical change due to the increase of temperature. In general changes are larger for the end of the century. However we can observe some changes of sign between the two periods for the same metrics.

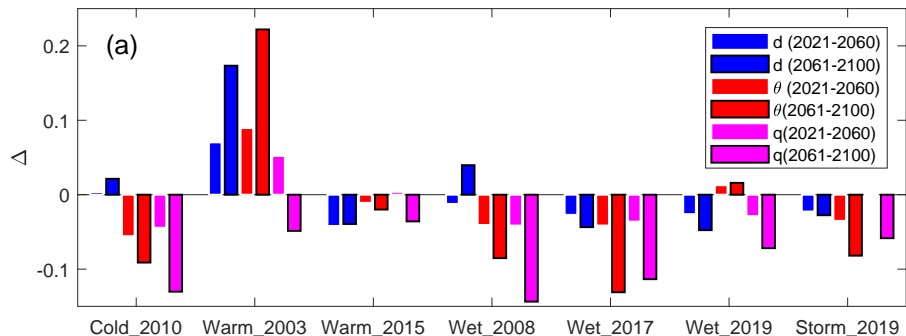

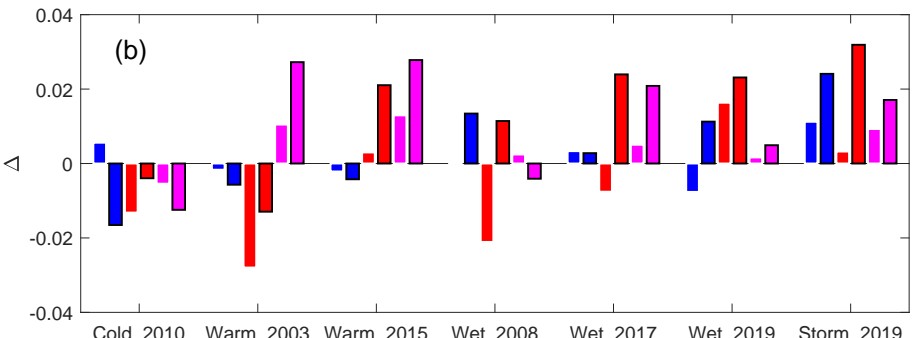

**Figure 10.** Upper panel (**a**): *d*, *θ* and analogue quality (*q*) changes in the "raw" simulations (Δ values). Lower panel (**b**): *d*, *θ* and analogue quality (*q*) changes in the "detrended" simulations. Each of the seven selected events is represented by six bars. The blue bars are for changes of local dimension *d* between a reference period 1979–2019 and two SSP585 scenario sub-periods (2021–2060 and 2061–2100). The red bars are for changes in persistence *theta*. The purple bars are for changes in the distances of Z500 analogues.

### 5.1. Circulation for Temperature Extremes

- Cold Spell 20210. From Figure 10, we see a modest change in the predictability, the persistence of this event increase in the future (negative variation of *θ*). The analogue quality decreases, meaning that the event is less probable in future climate scenarios, in terms of atmospheric circulation (not just temperature).
- Heatwave 2003. With the trend (Figure 10a) we have an increase of dimension (decrease of predictability) and decrease of persistence and small relative trends for the analogues quality. When removing the Z500 trend (Figure 10b), the dimension does not change, while the persistence increases and the event becomes more probable
- Warm winter 2015. The dynamical change (Figure 10b) has a similar amplitude as the thermodynamical change (Figure 10a). Therefore, the dynamical change is relatively large with respect to the thermodynamical change, in comparison with the other events.

### 5.2. Circulation for Precipitation Extremes

- Wet spring 2008 and wet winter 2017. We see no trend in the dimension *d*. The persistence increases, and the quality of analogues decreases (Figure 10a). When removing the Z500 trend, we get contrasting signal in the metrics, in particular with better analogues and thus a higher probability of occurrence of the spatial patterns.
- Wet Autumn 2019. The parameter changes with the raw data shows small negative trends in dimension and analogues quality (Figure 10a). The event become less persistent when removing the Z500 trend, and also more probable with an increased analogue quality (Figure 10b).

### 5.3. Circulation for a Wind Extreme

Similarly to the wet (and stormy) winter of 2017, the dynamical metrics changes yield opposite signs for the stormy winter of 2019: the values of dimension and persistence slightly (but consistently) decrease with the thermodynamical effect (due to warming) (Figure 10b), but the residual signal in dimension and persistence (due to potential pattern changes) increases with a similar amplitude (Figure 10b). This emphasises the importance of the dynamical signal (Figure 10b), with respect to the thermodynamical signal for the genesis of storms. The trends are enhanced between the first sub-period (2021–2060) and the second sub-period (2061–2100) of the CMIP6 SSP585 scenario simulations (blue and red bars in Figure 10).

## 6. Discussion and Conclusions

It has been argued that a major challenge for a complete understanding of climate extremes is the assessment of the atmospheric circulation leading to those events [7,8]. We have illustrated a statistical methodology to assess how the atmospheric circulation properties can change for emblematic recent events that hit France as the beginning of the 21st century, with a multi-model ensemble of climate simulations. This study provides an update of the analyses of Yiou et al. [62] and Faranda et al. [9] with CMIP6 [14] and ERA5 [15], who essentially used the CMIP5 database [63] and NCEP reanalysis [64] for attribution.

The results were obtained from the physical hypothesis that extreme values of temperature, precipitation or wind speed in France are linked to synoptic atmospheric circulation. This hypothesis is supported by various studies [13,65,66].

The raw trends in the dynamical metrics reflect a general increase of the Z500 values which depends on temperature. Indeed, the geopotential height increase as the column density increases and it can be directly connected to the increase in temperature caused by global climate change through the equation of state. This implies that the quality of analogues seem to increase for the summer, and decrease for the other seasons, as reflected by the theoretical study of Robin et al. [67] or the analyses of Vrac et al. [68] on seasonal changes of the atmospheric circulation. This corresponds to the "hammam" effect described by Faranda et al. [69]. This shows that the predictability of some events increases with climate change (although those extremes also become more probable).

Most of the signals conveying dynamical changes are rather small. This is due to the fact that the variability of the atmospheric circulation might be underestimated by climate models [70]. This is consistent with the results of Vautard et al. [71], who compared the relative contributions of thermodynamical and dynamical effects on the amplitude of extremely wet winters, although we report results for a large variety of events.

The trends of the dynamics in model simulations can even be opposite to the trend of reanalyses for stormy events [42]. This is the main caveat of our study, which relies on CMIP6 model simulations, whose spatial resolution is on average ten times coarser than the resolution of ERA5. On the other hand, taking a multi-model ensemble allows taking into account implicitly interdecadal variability due to oceans. We find that the changes are almost always amplified with the amplitude of climate change (from 2021–2060 to 2061–2100) when keeping the thermodynamic trend. This enhancement suggests the robustness of the results (which are not obtained at random). However, we cannot exclude that interdecadal variability affects the results when removing the trend.

Our results should be interpreted as changes in the likelihood of synoptic circulation (quality of analogues) and predictability of events ($d$-$\theta$) as a response to climate change, conditional to an emission scenario (SSP585) and a set of climate simulations (from CMIP6). The contrasting signals between the "raw" and "detrended" Z500 in CMIP6 illustrate the relative contributions of thermodynamical and dynamical effects in EEA. The orders of magnitudes are similar to what was found for a specific event (winter 2013/2014) [7,71,72], albeit with other datasets. We observe that the changes of dynamical indicators are 2.5% at most for wet events.

This (admittedly) technical approach to the analysis of extreme events and their relation with the large scale atmospheric circulation is a necessary step for *conditional attribution* (to the atmospheric circulation) of events [4]. We emphasise that this approach is very generic, and could be applied to other events, in other places of the world, see, e.g., [9,62].

**Author Contributions:** P.Y. selected the extreme events from ERA5. D.F. produced the $d$-$\theta$ diagnostics. M.V. contributed to the bias correction. S.T. contributed to the dataset management. All authors participated to the writing of the manuscript. All authors have read and agreed to the published version of the manuscript.

**Funding:** This research was funded by French ANR project No. ANR-20-CE01-0008 (SAMPRACE), the French Convention de Service Climatique (COSC), and the ERA4CS project EUPHEME. DF has been supported by the ANR Project No. ANR-19-ERC7-0003 (BOREAS) and by a LEFE-MANU-INSU-CNRS grant "DINCLIC". MV has been supported by the CoCliServ project, which is part of ERA4CS, an ERA-NET initiated by JPI Climate and cofunded by the European Union (Grant No. 690462). MV has also been supported by project C3S 428J ("HR-CDFt").

**Data Availability Statement:** ERA5 reanalysis data are available from the Copernicus platform (https://climate.copernicus.eu/climate-reanalysis) (last access on 29 October 2021). CMIP6 data (with bias correction) are available on the esgf platform at IPSL (https://esgf-node.ipsl.upmc.fr/) (accessed on 29 October 2021).

**Acknowledgments:** We thank Flavio Pons for his advice on data processing tools.

**Conflicts of Interest:** The authors declare no conflict of interest. The funders had no role in the design of the study; in the collection, analyses, or interpretation of data; in the writing of the manuscript; or in the decision to publish the results.

## Abbreviations

The following abbreviations are used in this manuscript:

| | |
|---|---|
| CMIP6 | Coupled Model Intercomparison Project phase 6 |
| SSP585 | Socio-economic Pathway No. 5 with 8.5 W·m$^{-2}$ forcing |
| Z500 | Geopotential height at 500 hPa |
| SLP | Sea-level pressure |
| $d$ | Local dimension |
| $\theta$ | Persistence |

## Appendix A. Quantification of Z500 Bias Correction

We show here the effect of bias correction of Z500 on the 11 CMIP6 simulations. We focus on two months (August and December), during which many of the selected events occur.

Figure A1 shows that the bias correction of Z500 actually removes all bias in the mean values of Z500.

Figure A2 shows how the variance of Z500 is affected by bias correction. The ratios of bias corrected Z500 to ERA5 Z500 is very close to 1.

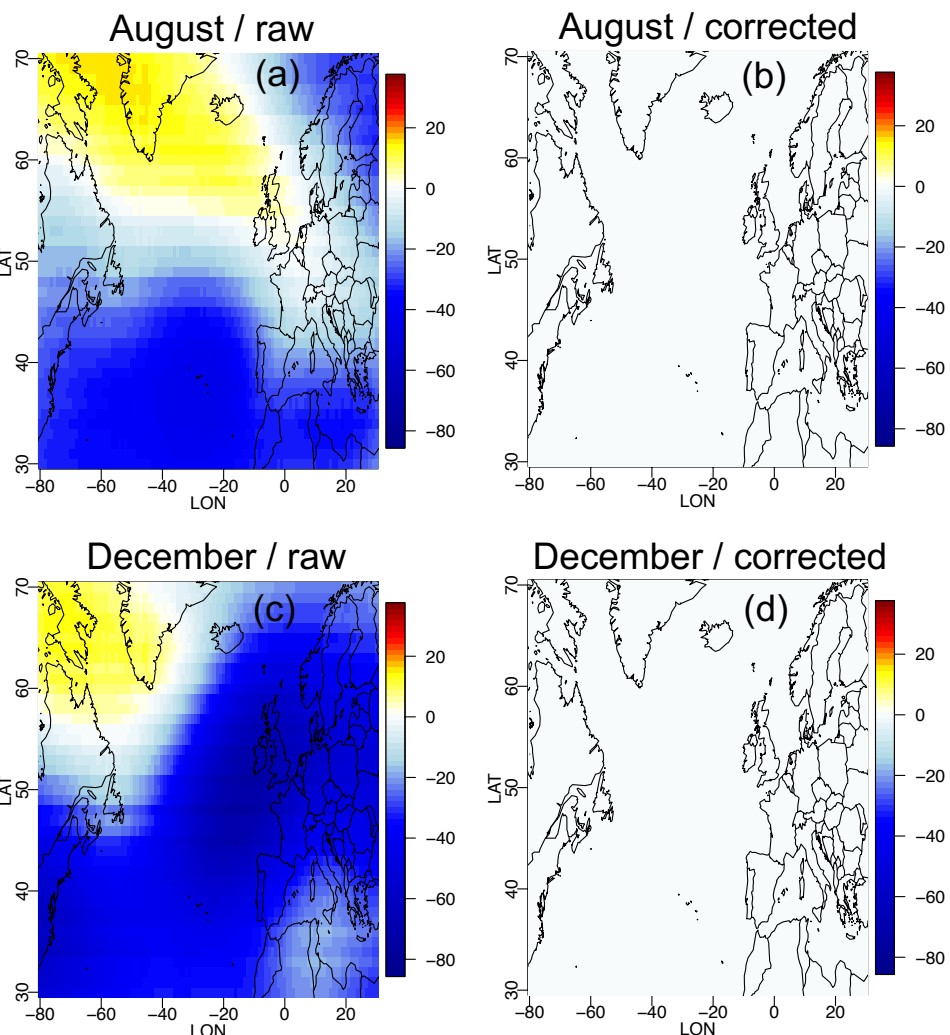

**Figure A1.** (**a**) Mean Z500 bias of 11 CMPI6 simulations (in m) for August. (**b**) Mean Z500 bias after bias correction for Z500 in August (in m). (**c**) Mean Z500 bias of 11 CMPI6 simulations (in m) for December. (**d**) Mean Z500 bias after bias correction for Z500 in December (in m).

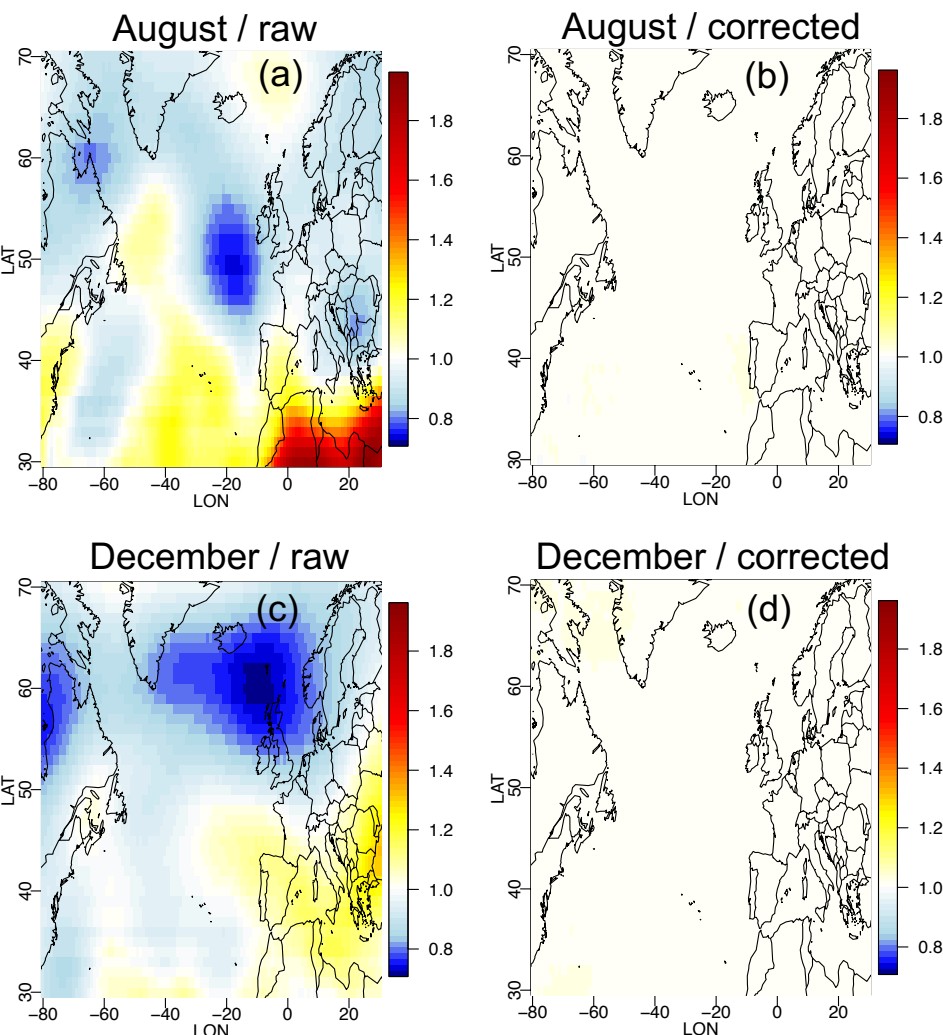

**Figure A2.** (**a**) Mean ratio of standard deviation of Z500 bias of 11 CMPI6 simulations for August. (**b**) Mean ratio of standard deviation of Z500 bias of 11 CMPI6 simulations after bias correction for August. (**c**) Mean ratio of standard deviation of Z500 bias of 11 CMPI6 simulations for December. (**d**) Mean ratio of standard deviation of Z500 bias of 11 CMPI6 simulations after bias correction for December.

### Appendix B. Distribution of $d$ and $\theta$ for CMIP6

We show the probability distributions of $d$ and $\theta$ for each selected event, among the CMIP6 (non detrended) Z500 data.

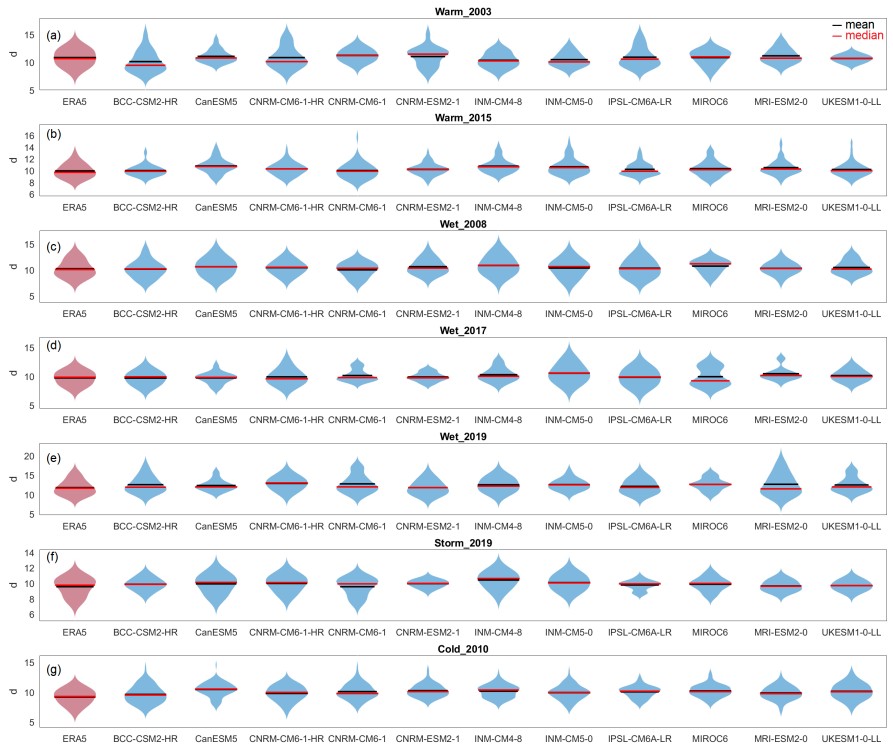

**Figure A3.** Each row shows violin boxplots of the daily distribution of *d* for selected extreme events. The red violins are for the ERA5 reanalysis. The other blue violins are for the 11 CMIP6 models. The horizontal black lines represent the mean. The horizontal red lines represent the median.

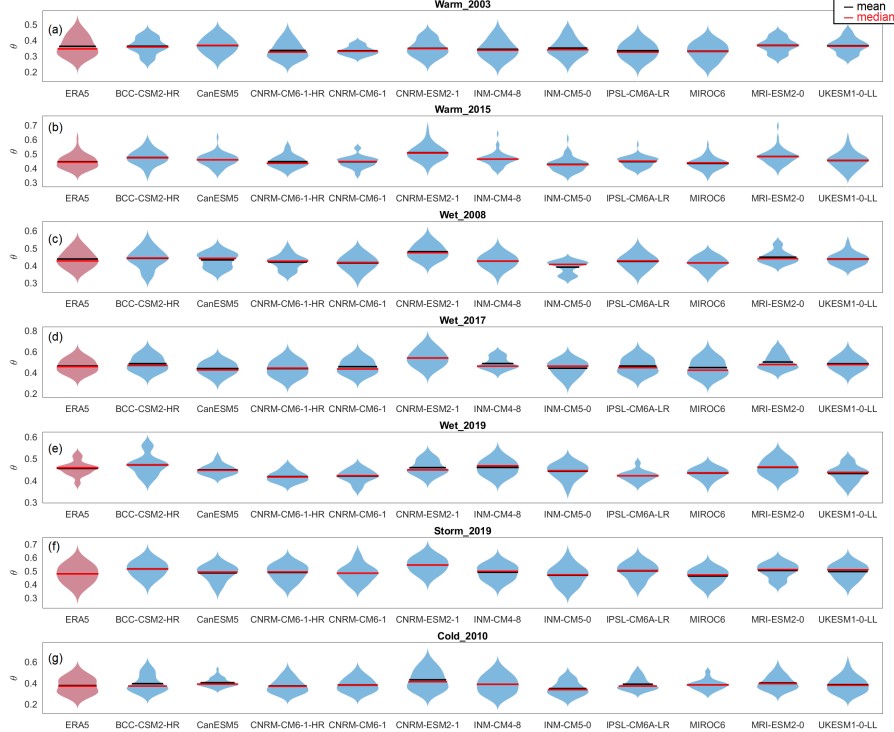

**Figure A4.** Each row shows violin boxplots of the daily distribution of *θ* for selected extreme events. The red violins are for the ERA5 reanalysis. The other blue violins are for the 11 CMIP6 models. The horizontal black lines represent the mean. The horizontal red lines represent the median.

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
