# Peer review of "Projected Changes in the Atmospheric Dynamics of Climate Extremes in France"

_atmosphere, doi:10.3390/atmos12111440_

Round 1

Reviewer 1 Report

I am happy to support publication of these results following some minor revisions.  Please see attached document for my detailed report.

Author Response

Comments to the author

Our replies are in blue.

The manuscript “Projected changes in the atmospheric dynamics of climate extremes in France" presents an analysis of recent extreme events in France and their representation in a future climate scenario. The authors take a focus on atmosphere circulation patterns (considering their probability, dimension, and persistence) due to their infuence on uncertainty in traditional extreme event attribution studies. Using the ERA5 reanalysis [2] as observations and a select sample of CMIP6 models [1] as the future scenario possibilities, they use tools from extreme event attribution which focus on dynamical properties related to prediction in order to explore potential changes in atmospheric circulation due to climate change.

While I feel the study is of high quality and relevant to the scope of the journal, I feel some improvements to the manuscript are necessary for publication. I offer some general comments below, as well as more specific points and technical corrections. Overall the work is very interesting and I would be happy to see it published if the authors address my comments.

We thank the reviewer for the positive appraisal of our paper.

General comments:

1. I feel that the two choices of analysis, i.e. \raw" vs \detrended", are not clearly motivated from the start. Lines 114-119 seem to give the main motivation, however it is lost in the way it follows the specifics of the debiasing and detrending procedures. Perhaps mentioning this earlier, even in the abstract and/or introduction section, would make the necessity of the two analyses much more straightforward.

This is now mentioned at the end of the introduction section.

2. The reasoning for choosing the particular extreme events studied here is not quite clear. The first part of the section discusses the probability of exceeding the extreme threshold on any given day. However according to Figure 2, most of the chosen events are not the years with the highest probability. The authors mention impacts of the various types of extreme events, but it is not clear if the events are chosen based on these impacts. I suggest clarifying and perhaps rewriting this section to better explain the connection between the probability measure and

the ultimate choice of events.

The criteria to choose the events was that (i) they occurred after 2000 and (ii) they maximized duration and spatial extent. This is what figure 2 reflects. The seven chosen events are the post 2000 maxima of each time series. The discussion on impacts comes after this “probability based” choice. This is clarified in the text. The section on event selection now appears earlier.

3. The discussion around analog quality is confusing throughout the manuscript. In the introduction it appears as though it is referred to as pattern likelihood. Then in section 2.4 it is defined as an average of Euclidean distances of the closest daily fields. I do not quite understand what this Euclidean distance is taken between, as it seems there are multiple Euclidean distances discussed. If I have misinterpreted this than perhaps it would be better to spell out the process

in a different manner. Finally, the variable q is used for analog quality in the results section but I do not believe it is ever formally defined.

This is now detailed by giving the definition of the Euclidean distance we use, and on which climate field.

4. In general, I think the discussion around the dynamical properties can be expanded. Since they are the primary quantitative measures for the results, it is important that they are thoroughly introduced and the physical implications of the different values explained. This could easily come naturally if the authors take my suggestion for section 2.4 below regarding subsections.

The section (now section 4.2) has been clarified by following the suggestion of the referee of splitting the discussion in subsections and introducing the physical implications of the indicators.

5. The authors make multiple statements regarding \low" vs.\high" dimension in discussing the extreme events, however in Figure 1 the extreme events seem to have dimension from 9-12. I am a bit confused about this figure in general, as it is not clear to me what the small dots represent (see specific comments below). More care needs to be taken in specifying the distinction between low and high dimension and the respective implications to the events of interest.

In the new version of the manuscript we do not refer anymore to the dimension of these events in section 3 but only for their changes (in the results section 4), so we have removed the sentences. We now believe that the manuscript is more linear.

6. It seems as though warming changes the dynamical properties of circulation patterns. Has this been discussed in previous works? It looks as though reference [3] has relevance here but is not connected to the results. It would be nice to have some discussion around this.

This is now mentioned in the discussion/conclusion.

Specific comments:

Line 58: No mention of what Section 3 covers here.

OK. Corrected.

Lines 85-86: How sensitive are the results of the study to using different ensemble members in the models?

Indeed, we only used one of the members of each ensemble. This caveat is now mentioned in section 2.2.

Line 88: What is the biased representation of Z500? Can this be briefly mentioned in the text for non-experts?

To quantify the Z500 biases of the 11 GCMs with respect to ERA5 and therefore the need to correct them, two new figures are drawn and given in the supplementary material section. The first figure (Fig. A1) shows the average (over the 11 GCMs) monthly mean biases of the models for August and December. In addition, it also includes the same information but computed from the bias-corrected GCMs. The second figure (Fig. A2) shows the average (over the 11 GCMs) monthly ratios of the modeled Z500 variance over the ERA5 Z500 variance. As in the previous figure, it also includes the same computations but from the bias-corrected GCMs.

Based on these two new figures, it is clear that GCMs have biases in their Z500 simulations and that bias correction allows to adjust correctly both means and variances.

Section 2.4: Adding separate subsections here for each dynamical property computed would make the entire section more reader-friendly.

The section 2.4 (now section 4.2) has been separated in few subsections as suggested.

Figure 1: This  figure seems out of place. It is not mentioned in the text at all until after Figure 2 is discussed. Perhaps move its location, or add a discussion of it somewhere earlier in the manuscript. Additionally, its not clear what the small colored dots represent. Are they all the days in the given event season? If so, I would think there would be according to the length of the seasons. Please explain how these are identified.

The section order was changed. Note this figure appears after the event definition. The discussion about low/high dimension persistence has been removed.

Lines 185-186: \each day d or year y" - should this be \each day d and year y" considering   is a function of both?

Corrected.

Line 187: Here it is mentioned Nseas is number of days in a season. Is this in each particular year, given that DJF can change between years? Maybe briefly state whether the actual year calendar dates are used or the data is on a uniform yearly grid.

Indeed, the number of days in a season varies from 90 (winter) to 92 (summer), with 91 days in winter leap years. The effective calculation reflects this slight variation. This is now mentioned in the manuscript.

Line 199: Should what is in parentheses be after \falls"? Also maybe change falls to autumns.

Falls are changed to Autumns. What appears in the parentheses refers to Spring. The Autumn flash floods have no documented agricultural impact (in France) and are relatively independent of river management, contrary to Winter or Spring floods.

Line 200: All other parentheses mention impacts while this one mentions cause. It would read better if the impacts were here instead.

The impacts are mentioned. A reference is given in the subsection that describes the event.

Lines 299-300: \the majority of the points for the cold spell" - what is the specific breakdown of high persistence and low dimension vs rest?

This sentence has been rewritten. A reference has been added.

Lines 301-302: \The 2019 wet event occupies a region characterized by blocking patterns" - this is not immediately obvious from the paper. Is there a reference for this?

This sentence has been rewritten. A reference has been added.

Lines 303-305: \one has to keep in mind that in summer the variance of the cloud of points is reduced around the median values" - why is this relevant?

Indeed, this is obscure. The sentence is removed.

Lines 318-321: \The impact of circulation changes is therefore smaller than the thermodynamical change due to the increase of temperature" - I feel like this is the major  finding of the manuscript, comparing the influence of thermodynamic changes on extremes to that of dynamical changes. I would suggest to clearly state this as a motivation of the paper either in the abstract and/or introduction.

OK. This is now mentioned in the introduction.

Line 330: Here it is stated that with trend the event has \almost zero signal", however the values are of the same magnitude as all of the events without the trend. I do not see how one is more significant than the other in that case.

Point taken. The dynamical change (from detrended Z500) has a similar amplitude as the thermodynamical change (from “raw” Z500). Therefore, the dynamical change is relatively large with respect to the thermodynamical change, in comparison with the other events. The manuscript is modified accordingly.

Lines 348-350: The explanation of thermodynamic being due to warming and dynamic due to potential pattern changes (I assume meaning changes in dynamics of the patterns?) is a good explanation but would be better placed much earlier in the manuscript when introducing the two different analyses.

See previous points. This is now mentioned in the introduction and the discussion sections.

Lines 356-358: \We illustrate a statistical methodology to assess how the atmospheric circulation properties can change for emblematic recent events that hit France since the beginning of the 21st century, with a multi-model ensemble of climate simulations." - As previously noted, the manuscript reads more like a comparison between the influence of thermodynamical changes and atmospheric

pattern changes on extreme events.

OK. See previous points. This is now mentioned in the introduction and the discussion sections, with added references.

Lines 365-366: It does not seem obvious how the trends in dynamical measures show a general increase in Z500 values. Does this mean warming implies higher pressure anomalies in general? Please elaborate on this.

We rephrased this sentence as follows: “The "raw" trends in the dynamical metrics reflect a general increase of the Z500 values which depends on temperature. Indeed the geopotential height increases as the column density increases and it can be directly connected to the increase in temperature caused by global climate change through the equation of state”.

Line 389: \rather weak" is a bit ambiguous. Can the authors use a more quantitative description or a comparison to the other changes?

The sentence has been rephrased: We observe that the changes of dynamical indicators are 2.5\% at most for wet events, while for other events changes can reach order of 4-5%

Technical corrections:

All done

Line 216: Reference missing

Line 241: Reference missing

Line 246: Reference missing

Line 315: \we" ! \We"

Line 364: \quite a few" ! \various"

Reviewer 2 Report

This paper used dimension d and persistence metric θ to represent the dynamical features of atmosphere for the seven selected events. The probability of atmospheric patterns and their predictability to respond to climate change was examined using CMIP6 models. I recommended this article be accepted after a minor modification.

1. How the effect of bias correction, authors should give the Z500 field for observation, raw and bias-corrected to demonstrate that it is necessary to conduct the bias correction. 
2. many sentences in this article are the same as that in reference 9. Authors should better rewrite those sentences.
3. delete a “the” in Line 171
4. The performance of CMIP6 models in simulating the d- θ parameters should be mentioned in the manuscript. 

Author Response

Reviewer #2

Our replies are in blue.

This paper used dimension d and persistence metric θ to represent the dynamical features of atmosphere for the seven selected events. The probability of atmospheric patterns and their predictability to respond to climate change was examined using CMIP6 models. I recommended this article be accepted after a minor modification.

Thank you for the positive appraisal of our paper.

1. How the effect of bias correction, authors should give the Z500 field for observation, raw and bias-corrected to demonstrate that it is necessary to conduct the bias correction.

To quantify the Z500 biases of the 11 GCMs with respect to ERA5 and therefore the need to correct them, two new figures are drawn and given in the supplementary material section. The first figure (Fig. A1) shows the average (over the 11 GCMs) monthly mean biases of the models for August and December. In addition, it also includes the same information but computed from the bias-corrected GCMs. The second figure (Fig. A2) shows the average (over the 11 GCMs) monthly ratios of the modeled Z500 variance over the ERA5 Z500 variance. As in the previous figure, it also includes the same computations but from the bias-corrected GCMs.

Based on these two new figures, it is clear that GCMs have biases in their Z500 simulations and that bias correction allows to adjust correctly both means and variances.

2. many sentences in this article are the same as that in reference 9. Authors should better rewrite those sentences.

The paper of Faranda et al. (GRL, 2020) was rather methodological. The present paper applies the methodology defined in that paper to other case studies (and other datasets). We tried to attenuate the resemblance, but there is no fundamental reason for paraphrasing what we consider is well formulated. No sentence in the results or conclusion sections is replicated from other papers.

3. delete a “the” in Line 171

Done.

4. The performance of CMIP6 models in simulating the d- θ parameters should be mentioned in the manuscript.

We added two figures (A3 and A4) that show the d-theta distribution for each event and each model simulation.